# Learning Latent State Spaces for Planning through Reward Prediction

## Abstract

Model-based reinforcement learning methods typically learn models for high-dimensional state spaces by aiming to reconstruct and predict the original observations. However, drawing inspiration from model-free reinforcement learning, we propose learning a latent dynamics model directly from rewards. In this work, we introduce a model-based planning framework which learns a latent reward prediction model and then plan in the latent state-space. The latent representation is learned exclusively from multi-step reward prediction which we show to be the only necessary information for successful planning. With this framework, we are able to benefit from the concise model-free representation, while still enjoying the data-efficiency of model-based algorithms. We demonstrate our framework in multi-pendulum and multi-cheetah environments where several pendulums or cheetahs are shown to the agent but only one of them produces rewards. In these environments, it is important for the agent to construct a concise latent representation to filter out irrelevant observations. We find that our method can successfully learn an accurate latent reward prediction model in the presence of the irrelevant information while existing model-based methods fail. Planning in the learned latent state-space shows strong performance and high sample efficiency over model-free and model-based baselines.

## 1 Introduction

Deep reinforcement learning (DRL) has demonstrated some of the most impressive results on several complex, high-dimensional sequential decision-making tasks such as video games (Mnih et al., 2015) and board games (Silver et al., 2018). Such algorithms are often model-free, i.e. they do not learn or exploit knowledge of the environment's dynamics. Although these model-free schemes can achieve state-of-the-art performance, they often require millions of interactions with the environment. The high sample complexity restricts their applicability to real environments, where obtaining the required amount of interactions is prohibitively time-consuming. On the other hand, model-based approaches can achieve low sample complexity by first learning a model for the environment. The learned model can then be used for planning, thereby minimizing the number of environment interactions. However, model-based methods are typically limited to low-dimensional tasks or settings with certain assumptions imposed on the dynamics (Li & Todorov, 2004; Deisenroth & Rasmussen, 2011; Levine & Koltun, 2013; Chebotar et al., 2017)

Several recent works have learned dynamics models parameterized by deep neural networks (Lenz et al., 2015; Finn & Levine, 2017; Williams et al., 2017), but such methods generally result in high-dimensional models which are unsuitable for planning. In order to efficiently apply model-based planning in high-dimensional spaces, a promising approach is to learn a low-dimensional latent state representation of the original high-dimensional space. A compact latent representation is often more conducive for planning, which can further reduce the number of environment interactions required to learn a good policy (Watter et al., 2015; Banijamali et al., 2017; Finn et al., 2016; Ichter & Pavone, 2019; Zhang et al., 2019; Hafner et al., 2019). Typically, such a model has a variational autoencoder (VAE) (Kingma & Welling, 2013)-like structure that learns a latent representation. In particular, it learns this representation through reconstruction and prediction of the full-state from the latent state. In other words, the learned low-dimensional latent state must retain a sufficient amount of information from the original state to accurately predict the state in the high-dimensional space. Although the ability to perform full-state prediction is sufficient for planning in latent space, it is not

strictly necessary. Predicting the full-state may result in a latent state representation which contains information irrelevant to planning. For example, consider a task of maze navigation, with a TV placed in the maze, displaying images (Burda et al., 2018a). The TV content is irrelevant to the task, but the latent representation may learn features designed to predict the state of the TV.

In RL, the objective is to maximize the cumulative sum of rewards. Given this objective, it may be excessive to learn to perform full-state prediction, if aspects of the full state have no influence on the reward. In examples like the aforementioned TV in a maze, it may be more fruitful to instead optimize a latent model to be able to perform reward prediction. In some sense, value-based model-free methods can be seen as reward prediction models. However, they attempt to predict the optimal value which is policy dependent and requires a large number of samples to learn. Fortunately, model-based planning methods such as model predictive control (MPC) only maximize the cumulative sum of future rewards over the choices of action sequences. This implies that what a planning method requires is a latent reward prediction model: a model which predicts current and future rewards over a horizon from a latent state under different action sequences. In this work, we introduce a model-based DRL framework where we learn a latent dynamics model exclusively from the multi-step reward prediction criterion and then use MPC to plan directly in the latent state-space. Learning the latent reward prediction model benefits from the sample-efficiency of model-based methods, and the learned latent model is more useful for planning than its full-state prediction counterparts as discussed above.

The contributions of this paper are as follows. The proposed latent model is learned only from multi-step reward prediction. Optimizing exclusively for reward prediction allows us to circumvent the inefficiency stemming from learning irrelevant parts of the state space, while maintaining the model-based sample efficiency. We provide a performance guarantee on planning in the learned latent state-space without assumptions on the underlying task dynamics. Empirically, we demonstrate that our method outperforms model-free and model-based baselines on multi-pendulum and multi-cheetah environments where the agent must ignore a large amount of irrelevant information.

The remainder of this paper is structured as follows. In Section 2, we present related work in model-based DRL. In Section 3, we provide the requisite background on RL and MPC. In Section 4, we introduce our method. In Section 5, we present our experimental results, and conclude by indicating potential future directions of research.

## 2 RELATED WORK

**Model-based DRL**    Model-based works have aimed to increase sample efficiency and stability by utilizing a model during policy optimization. Methods like PILCO (Deisenroth & Rasmussen, 2011), DeepMPC (Lenz et al., 2015), and MPPI (Williams et al., 2017) have exhibited promising performance on some control tasks using only a handful of episodes, however they usually have difficulty scaling to high-dimensional observations. Guided-policy search methods (Levine & Koltun, 2013; Chebotar et al., 2017) make use of local models to update a global deep policy, however these local assumptions could fail to capture global properties and suffer at points of discontinuity. For high-dimensional observations such as images, deep neural networks have been trained to directly predict the observations (Finn & Levine, 2017), but the resulting high-dimensional models limits their applicability in efficient planning.

**Learning latent state spaces for planning**    To make model-based methods more scalable, several works have adopted an auto-encoder scheme to learn a reduced-dimensional latent state representation. Then one may plan directly in the latent space using some conventional planning methods such as linear quadratic regulator (LQR), or rapidly exploring random tree (RRT) (Watter et al., 2015; Banijamali et al., 2017; Finn et al., 2016; Ichter & Pavone, 2019; Zhang et al., 2019). However, these methods construct latent spaces based on state reconstruction/prediction which may not be entirely relevant to the planning problem as discussed in the previous section. A recent work PlaNet (Hafner et al., 2019) offered a latent space planning approach by predicting multi-step observations and rewards. The PlaNet has shown to achieve good sample complexity and performance compared to the state-of-the-art model-free methods, but its latent model might also suffer from the aforementioned issues with irrelevant observation predictions. In this work we use a similar planning algorithm to PlaNet, however, removing the dependency of observation prediction provides a more concise latent representation for planning.

**Reward-based representations learning for RL** The work of VPN (Oh et al., 2017) proposed a RL framework which learns a model to predict the future values rather than the full-states of the environment. Although VPN shares a very similar idea as our framework to not predict irrelevant information in full-state, it suffers the sample complexity issue of learning the optimal value as in model-free methods. Performance guarantees of latent models were analyzed under the DeepMDP framework in a recent paper (Gelada et al., 2019) without full-state reconstruction or prediction. With Lipschitz assumptions, their work provides performance bounds for latent representations when the two prediction losses of the current reward and the next latent state are optimized. They also show improvements in model-free DRL by adding the DeepMDP losses as auxiliary objectives. However, with only the single-step prediction loss, DeepMDP may not be suitable for making long term reward predictions necessary for planning algorithms. In addition, the latent state prediction loss may be unnecessary and could result in local minima observed in their paper. Our framework shares the same spirit of DeepMDP by learning latent dynamics only from the reward sequences, but focusing on multi-step reward prediction allows us to achieve higher performance in model-based planning.

## 3 Preliminaries

### 3.1 Problem Setup

In this paper we consider a discrete time nonlinear dynamical system $f : \mathcal{S} \times \mathcal{A} \to \mathcal{S}$ with continuous state and action spaces $\mathcal{S} \subseteq \mathbb{R}^{d_n}$, $\mathcal{A} \subseteq \mathbb{R}^{d_m}$, and a reward function $R : \mathcal{S} \times \mathcal{A} \to \mathbb{R}$. Then, given an admissible action $a_t \in \mathcal{A}$ at time $t$, the system state $s_t$ evolves according to the dynamics

$$s_{t+1} = f(s_t, a_t) \tag{1}$$

where $s_{t+1}$ is the next state at time $t + 1$.

Our goal is to find a policy $\pi : \mathcal{S} \to \mathcal{A}$ that selects actions $\{a_t = \pi(s_t), t = 0, 1, \ldots\}$ so as to maximize the cumulative discounted rewards

$$\sum_{t=0}^{\infty} \gamma^t R(s_t, a_t) \tag{2}$$

where the initial state $s_0$ is assumed to be fixed, and $\gamma$ is the discount factor.

Note that Equations (1) and (2) formulate an infinite horizon optimal control problem. This problem can also be viewed as a deterministic Markov decision process (MDP) by the tuple $(\mathcal{S}, \mathcal{A}, f, R, \gamma)$.

### 3.2 MPC Planning

When a dynamics model is available, model predictive control (MPC) is a powerful planning framework for optimal control problems. A MPC agent chooses its action by online optimization over a finite planning horizon $H$. More specifically, at each time $t$, the agent solves a $H$-horizon trajectory optimization problem:

$$\max_{\hat{a}_{t:(t+H-1)}} J(s_t, \hat{a}_{t:(t+H-1)}) \tag{3a}$$

$$\text{subject to} \quad J(s_t, \hat{a}_{t:(t+H-1)}) = \sum_{\tau=t}^{t+H-1} \gamma^{\tau-t} R(\hat{s}_\tau, \hat{a}_\tau) \tag{3b}$$

$$\hat{s}_t = s_t, \text{ and } \hat{s}_{\tau+1} = f(\hat{s}_\tau, \hat{a}_\tau) \text{ for } \tau = t, t+1, \ldots, (t+H-2) \tag{3c}$$

Suppose the optimal control sequence is $\hat{a}^*_{t:(t+H-1)}$, then the MPC agent will select $\hat{a}^*_t$ as the control action at time $t$. This method is termed model predictive control due to its use of the model $f$ to predict the future trajectory $\hat{s}_{(t+1):(t+H-1)}$ from state $s_t$ and the intended action sequence $\hat{a}_{t:(t+H-1)}$. Let $\pi_{\text{MPC}}(s_t|f, R)$ denote the MPC policy using the dynamics model $f$ and reward $R$. Then the MPC agent selects its action by $\pi_{\text{MPC}}(s_t|f, R) = \hat{a}^*_t$.

### 3.3 Sample-Based Trajectory Optimization

From the description in Section 3.2, an MPC agent's need to solve the trajectory optimization problem specified by Equation equation 3. In this work, we use the cross entropy method (CEM) (Rubinstein,

1997) to solve problem equation 3. CEM is a very simple sampling-based planning method which only relies on a forward model and trajectory returns which may make direct use of our learned latent mappings somewhat like a simulator. The advantage to using a method like CEM, is that it's sampling procedure can be easily parallelized directly in the learned latent space.

## 4 LATENT REWARD PREDICTION MODEL FOR PLANNING

In order to perform model-based planning directly from observations without a model, we need to learn a dynamics model. For high-dimensional problems, a common model-based DRL approach is to learn a latent representation and then conduct planning in the low-dimensional latent space. In this work, rather than predict the full state/observation, we learn a latent state-space model which predicts only the current and future rewards conditioned on action sequences. We will discuss that observation reconstruction is not necessary if we can predict the rewards well over the planning horizon.

### 4.1 LEARNING A LATENT REWARD PREDICTION MODEL

Our model consist of three distinct components which can be trained in an end-to-end fashion. To do so, we first embed the state observation $s_t$ to the latent state $z_t$ with the parameterized function $\phi_\theta : \mathcal{S} \to \mathcal{Z} \subseteq \mathbb{R}^{d_z}$. In order to propagate the latent state forward in latent space, we define the forward dynamics function which maps $f_\psi^z : \mathcal{Z} \times \mathcal{A} \to \mathcal{Z}$, corresponding to the discrete-time dynamical system in latent state space. Finally, we require a reward function $R_\zeta^z : \mathcal{Z} \times \mathcal{A} \to \mathbb{R}$ which provides us an estimate of the reward given a latent state $z_t$ and control action $a_t$. The full latent reward prediction model is depicted in Figure 1.

$$z_t = \phi_\theta(s_t), \quad \hat{z}_{t+1} = f_\psi^z(z_t, a_t), \quad \hat{r}_t = R_\zeta^z(z_t, a_t) \tag{4}$$

Note that we do not require a "decoding" function since planning algorithms often only need to evaluate the rewards along a trajectory given an action sequence.

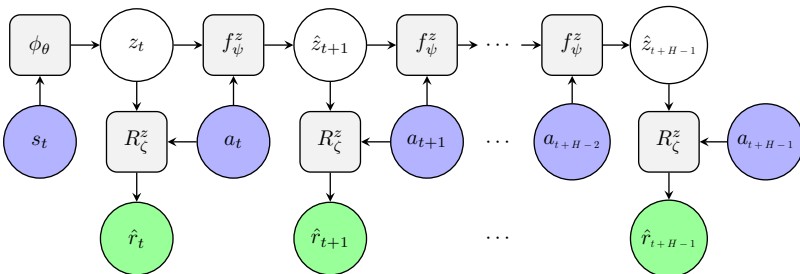

Figure 1: The latent reward prediction model $(\phi_\theta, f_\psi^z, R_\zeta^z)$. Blue and green circles represent the input and output variables. The squares indicate components of the prediction model and white circles represent the latent variables. The model takes in an initial state $s_t$ and a sequence of actions $a_{t:(t+H-1)}$ to return a sequence of $H$-step predicted rewards $\hat{r}_{t:(t+H-1)}$.

To achieve the aforementioned reward prediction ability, we learn these functions $\phi_\theta$, $f_\psi^z$, $R_\zeta^z$ over multi-step reward prediction losses. Formally, given a $H$-step sequence of states and actions $\{(s_k, a_k), k = t, \ldots, t + H - 1\}$, the training objective is defined to be the mean-squared error between the original rewards and the multi-step latent reward predictions.

$$\mathcal{L}_H = \frac{1}{H} \sum_{\tau=t}^{t+H-1} ||\gamma^{\tau-t} R(s_\tau, a_\tau) - \gamma^{\tau-t} R_\zeta^z(\hat{z}_\tau, a_\tau)||_2^2 \tag{5}$$

$$\text{where} \quad \hat{z}_\tau = f_\psi^z(\ldots f_\psi^z(\phi_\theta(s_t), a_t), \ldots, a_{\tau-1}) \tag{6}$$

We later show that this $H$-step reward prediction loss is sufficient to bound the planning performance over the same horizon.

Note that we do not impose other losses such as the observation prediction loss or latent state prediction loss as in some previous latent model frameworks (Gelada et al., 2019; Hafner et al., 2019).

The reason is that other loss functions are not necessary for planning performance. Having those additional losses would increase the training complexity and may induce undesirable local minima.

## 4.2 MPC with the Latent Reward Prediction Model

Once a latent state-space model is learned, we can perform MPC using the latent embedding. Let $\pi_{\text{MPC}}^z(s_t|\phi_\theta, f_\psi^z, R_\zeta^z)$ be the latent MPC policy with the latent reward model $(\phi_\theta, f_\psi^z, R_\zeta^z)$. Similar to Equation (3), the latent MPC agent at each time solves a $H$-step trajectory planning problem.

$$\max_{\hat{a}_{t:(t+H-1)}} J^z(s_t, \hat{a}_{t:(t+H-1)}) \tag{7a}$$

$$\text{subject to} \quad J^z(s_t, \hat{a}_{t:(t+H-1)}) = \sum_{\tau=t}^{t+H-1} \gamma^{\tau-t} R_\zeta^z(\hat{z}_\tau, \hat{a}_\tau) \tag{7b}$$

$$\hat{z}_t = \phi_\theta(s_t), \text{ and } \hat{z}_{\tau+1} = f_\psi^z(\hat{z}_\tau, \hat{a}_\tau) \text{ for } \tau = t, t+1, \ldots, (t+H-2) \tag{7c}$$

After solving the optimization problem, the latent MPC agent will choose its action by

$$\pi_{\text{MPC}}^z(s_t|\phi_\theta, f_\psi^z, R_\zeta^z) = \hat{a}_t^* \tag{8}$$

$$\text{where } \hat{a}_{t:(t+H-1)}^* = \underset{\hat{a}_{t:(t+H-1)}}{\text{argmax}} J^z(s_t, \hat{a}_{t:(t+H-1)}) \tag{9}$$

Similar to other model-based methods, the latent reward prediction model can be used in an offline setting with a pre-collected dataset. Together with an exploration policy, e.g. $\epsilon$-greedy, our model can also be used in an online iterative scheme. See appendix for the descriptions of Algorithm 1 (offline) and Algorithm 2 (online iterative).

## 4.3 Planning Performance Guarantee

To evaluate the performance of MPC planning, define recursively the optimal $n$-step Q-function by

$$Q_1^*(s, a) = R(s, a),$$
$$Q_n^*(s, a) = R(s, a) + \gamma \max_{a' \in \mathcal{A}} Q_{n-1}^*(f(s, a), a') \text{ for } n = 2, 3, \ldots, H$$

Then the Bellman optimality shows that $Q_n^*(s, a)$ gives the optimal value for the MPC planning problem (3), i.e., $\max_{\hat{a}_{t:(t+H-1)}} J(s_t, \hat{a}_{t:(t+H-1)}) = \max_a Q_H^*(s_t, a)$.

Similarly, define the optimal $n$-step Q-functions for the latent state-space model $(\phi_\theta, f_\psi^z, R_\zeta^z)$:

$$Q_1^{*,z}(z, a) = R_\zeta^z(z, a),$$
$$Q_n^{*,z}(z, a) = R_\zeta^z(z, a) + \gamma \max_{a' \in \mathcal{A}} Q_{n-1}^{*,z}(f_\psi^z(z, a), a')) \text{ for } n = 2, 3, \ldots, H$$

We have the following result connecting the $H$-step reward prediction loss and the $H$-step Q-functions.

**Theorem 1.** *Suppose the $H$-step prediction losses $\mathcal{L}_H \leq \epsilon^2$ for any $H$-step trajectory. Then the $H$-step Q-functions satisfy*

$$Q_H^{*,z}(\phi_\theta(s), a) \geq Q_H^*(s, a) - \epsilon H^{1/2}$$

*for all $s \in \mathcal{S}, a \in \mathcal{A}$. Moreover, if $\pi_{MPC}^z(s|\phi_\theta, f_\psi^z, R_\zeta^z)$ is optimal for the latent MPC planning problem equation 7, then it is also a $(2\epsilon H^{1/2})$-optimal policy for the MPC planning problem (3).*

This result provide a planning performance guarantee for the latent reward prediction model. It also confirms that we don't need other unnecessary losses to achieve the desired performance from the latent model. Note that this result does not require additional assumptions such as Lipschitz continuity of the environment. See the appendix for a proof of the theorem.

## 5 EXPERIMENTS

In this section we introduce several experiments and their results which are aimed to investigate the representation ability, long-term prediction ability and sample efficiency. We implement the mappings $\phi_\theta, f_\psi^z, R_\zeta^z$ by deterministic feed-forward neural networks. The extensions to stochastic and/or recurrent neural networks are potential future directions. In order to do MPC, we choose to use CEM with a planning horizon of $H = 12$ and $K = 1000$ trajectory samples. We use the $k = 100$ best trajectories to compute the control, re-planning after every action. CEM will use the model somewhat like a simulator to collect samples in latent space (in our case this is a batch forward pass in a deep network). We consider two sets of environments with high-dimensional irrelevant observations. Results for an additional image-based environment is also available in the appendix.

### 5.1 BASELINES

Before introducing the experiments, we briefly introduce the baseline algorithms used for comparison. Throughout our experiments, we have several baselines, each serving to demonstrate the attributes of our reward model: concise representation, long-term prediction and sample efficiency.

**State prediction model**   To demonstrate the representation ability of our reward prediction model, we consider a latent model where the latent space is learned from state prediction only. We train a reward predictor separately for the model. This model works well for the concise representations, however, like many model-based methods it fails to scale to high-dimensional observations.

**DeepMDP**   We use the DeepMDP model (Gelada et al., 2019), which is essentially a one-step variant of the reward prediction model with an additional latent state prediction loss. We aim to show that our multi-step reward prediction-only loss improves the planning performance.

**SAC**   Finally to demonstrate sample efficiency, we compare against the model free Soft Actor-Critic algorithm (SAC) (Haarnoja et al., 2018). SAC is currently a popular DRL algorithm known for its sample efficiency and performance.

### 5.2 MULTI-PENDULUM

In several RL tasks, there may be irrelevant aspects of the state, e.g. the noisy TV problem (Burda et al., 2018b), so reward prediction might be extremely crucial to learn a more conducive and useful model for planning. To verify this claim, we develop a multi-pendulum environment, an adaptation of the classic pendulum control task (Brockman et al., 2016), where the agent must swing up and stabilize a pendulum at the up-right state with torque constraints. Although this task is simple, it encompasses nonlinear dynamics and long planning horizons in order to accumulate enough energy to swing up. In the multi-pendulum task, the agent operates in an environment with $N$ pendulums, but is only rewarded for swinging up a single pendulum. The other $N - 1$ pendulums serving as determined observational noise being driven by uniform random actions. The state observation for the $N$-pendulum environment is given as:

$$s = [\sin(\theta_1), \cos(\theta_1), \dot{\theta}_1, \sin(\theta_2), \cos(\theta_2), \dot{\theta}_2, \ldots, \sin(\theta_N), \cos(\theta_N), \dot{\theta}_N] \in \mathbb{R}^{3N}$$

Since reward prediction model can be used either completely offline or iterative online setting, we present two sets of experiments. First we show the average final performances of Algorithm 1 and baselines after training completely offline from 20000 steps of the environment under time-correlated noised control. We vary the number of pendulums $N \in \{1, 2, 5, 10, 15\}$ where we maintain the latent space to be 3-dimensional throughout. In the second experiment we train the model and baselines under the iterative scheme described in Algorithm 2, consuming approximately the same number of samples. We also compare against the SAC algorithm as a state-of-the-art model-free baseline.

As shown in Figure 2, the reward prediction loss does not degrade quickly for our reward prediction model as the number of irrelevant pendulums increases, but degrades almost immediately for the state prediction model. DeepMDP struggles to predict well in all cases, due to only training on a single-step latent plus reward reconstruction loss. In Table 1, we can further see that performance of the reward prediction model (Algorithm 1) is not very sensitive to the number of excess pendulum

environments, where the state prediction model immediately fails to solve the task. Note that the SAC method requires a significantly greater number of samples to achieve similar performance.

For the iterative scheme (Algorithm 2), the reward prediction model clearly outperforms all baselines as can be seen in Figure 3. Note that the DeepMDP model has a much higher variance than other methods. This high variance could be due the local optima from the latent state prediction loss in DeepMDP. On the other hand, our reward prediction model provides a stable low-variance result.

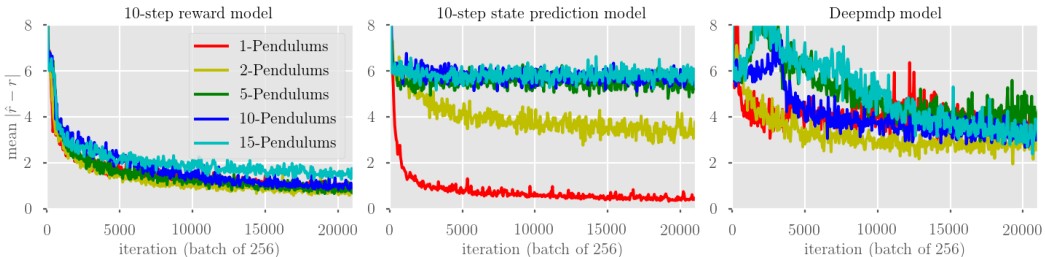

Figure 2: Training curves for the 10-step reward prediction loss. All experiments are ran in the offline setting using 20000 steps from the true environment under a random policy and trained for 300 epochs. The results of our reward prediction model is shown in the left, the state prediction model in the middle, and DeepMDP in the right.

| | Reward Model | State Model | DeepMDP | SAC ($10^6$ samples) |
|---|---|---|---|---|
| # pendulums | mean(std) | mean(std) | mean(std) | mean(std) |
| 1 | $-138.28(103.83)$ | $\mathbf{-137.65(79.01)}$ | $-374.25(113.42)$ | $-145.26(92.10)$ |
| 2 | $\mathbf{-163.99(79.76)}$ | $-451.72(188.01)$ | $-373.466(146.48)$ | $-150.31(90.53)$ |
| 5 | $\mathbf{-166.178(79.76)}$ | $-858.10(175.40)$ | $-665.09(196.32)$ | $-151.85(100.82)$ |
| 10 | $\mathbf{-156.09(91.46)}$ | $-912.63(160.35)$ | $-539.85(130.81)$ | $-151.41(83.21)$ |
| 15 | $\mathbf{-185.36(128.84)}$ | $-922.45(151.76)$ | $-578.89(172.26)$ | $-157.92(94.21)$ |

Table 1: Statistics over the final performance of the multi-pendulum environment trained under the offline setting from 20000 environment steps under a random-policy (excluding SAC). For each of the latent model, we use CEM for MPC planning, and each method is evaluated from 100 episodes.

## 5.3 MULTI-CHEETAH

In order to demonstrate that our method scales to higher dimensional environment, we use the Deepmind control suite "cheetah-run" environment (Tassa et al., 2018) to construct "multi-cheetah". The cheetah-run environment has a continuous 17-dimensional observation space and 6-dimensional action space. Similar to multi-pendulum, we concatenate each cheetah observation vector together into one observation, where only the first cheetah environment return the reward signal. Because a random policy is not sufficient to explore the state space in cheetah, the remaining $N-1$ cheetahs act according to an expert SAC policy. In multi-cheetah, we mainly compare our method with the model-free baseline SAC to highlight the strong performance and sample efficiency. Shown in figure 4, the reward prediction model is able to learn and plan near-optimally using far less samples than SAC in the 1-cheetah environment (i.e., the regular cheetah-run). Within 500 episodes, the reward model is able to achieve a near-optimal running behavior (503 and 501 final return in evaluation for 1-cheetah and 5-cheetah respectively), which takes SAC 3000 episodes to achieve. After 1000 episodes of 1-cheetah (not shown in the figure), the reward model achieves 570 return, approaching SAC's maximum performance of 590 after 4000 episodes. When there are several irrelevant running cheetahs, the performance SAC further drops. In comparison, our reward prediction model is rather agnostic to the irrelevant observations and maintains high performance and sample efficiency.

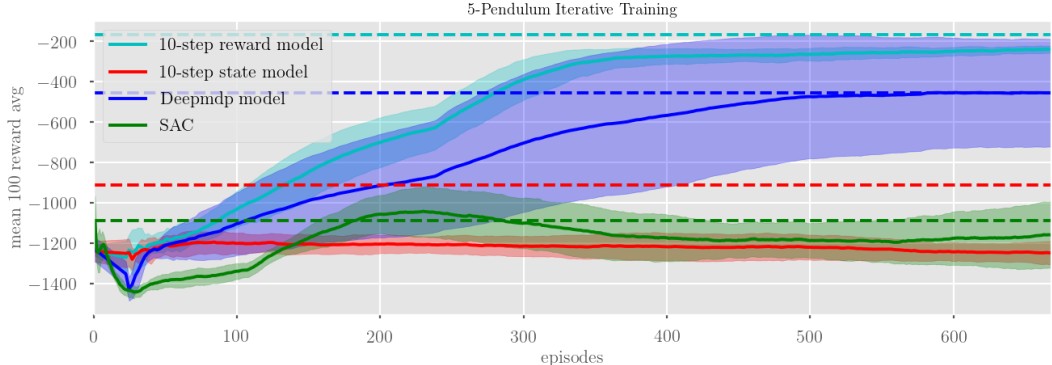

Figure 3: Training curves for the iterative scheme in the 5-pendulum environment. We initialize the each model with 2500 steps, and then perform 100 training iterations of batch-size 256 after each episode is collected with the current policy with $\epsilon = 0.7$-greedy exploration. Each method is run with 5 different seeds. The solid lines and shaded areas are the average training returns and their one standard deviation regions under the exploration policy. The corresponding dashed lines mark the average final evaluation performance after $\approx 23800$ environment samples.

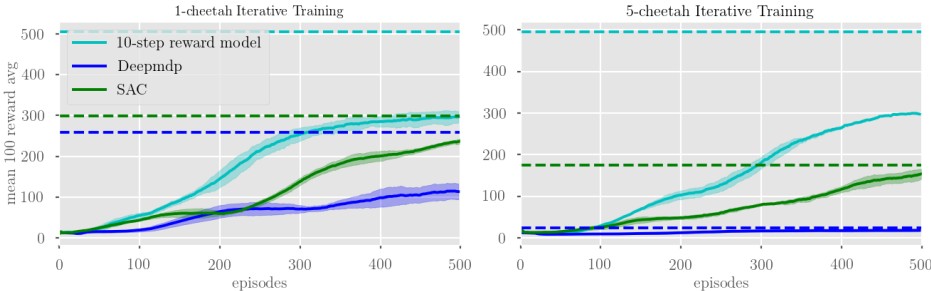

Figure 4: The average return training curves for SAC, Deepmdp and our reward prediction model in the 1-cheetah and 5-cheetah environments (85-dimensional). The reward model is trained iteratively online with zero-mean Gaussian exploration noise. Solid lines indicate training return (with 5 seeds and shaded one standard deviation regions), and the dashed lines are the final evaluation returns without exploration noise.

## 6 CONCLUSION

In this paper we introduced a method for learning a latent state representation from rewards. By constructing a latent space based on multi-step reward predictions, we obtained a concise representation similar to that of model-free DRL algorithms while maintaining high sample efficiency by planning with model-base algorithms. In this work we have used a sample-based CEM algorithm for MPC planning, but the use of other planning algorithms is also plausible. We demonstrated in the multi-pendulum and multi-cheetah environments that the latent reward prediction model is able to succeed in the presence of high-dimensional irrelevant information with both offline and iterative online schemes.

Although, there are several aspects which still need to be investigated such as more complex environments and sparse reward settings, where typically a finite-horizon planner would struggle. We would also like to investigate how to incorporate model uncertainty through either stochastic dynamics or ensemble methods. One interesting question we have is what is the optimal observer or what the optimal combination of reward and state reconstruction to do well in the task. We believe our minimal formulation is a good starting point for this question.

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

# A   ALGORITHM DESCRIPTIONS

---

**Algorithm 1** Offline Training

---

Inputs: planning horizon $H$, dataset $D$, latent model $(\phi_\theta, f_\psi^z, R_\zeta^z)$.

Train $(\phi_\theta, f_\psi^z, R_\zeta^z)$ using the loss function $\mathcal{L}_H$ in equation 5 and data $D$.

Output: policy $\pi_{\text{MPC}}^z(s_t|\phi_\theta, f_\psi^z, R_\zeta^z)$

---

---

**Algorithm 2** Online Iterative Scheme

---

Inputs: planning horizon $H$, number of iterations $N$, latent model $(\phi_\theta, f_\psi^z, R_\zeta^z)$.

Initialize dataset $D$ with random trajectories.

**for** $i = 1$ **to** $N$ **do**

    Train $(\phi_\theta, f_\psi^z, R_\zeta^z)$ using the loss function $\mathcal{L}_H$ in equation 5 and data $D$.

    Run latent MPC $\pi_{\text{MPC}}^z(s_t|\phi_\theta, f_\psi^z, R_\zeta^z)$ in the environment with $\epsilon$-greedy and collect data $D_i$.

    $D \leftarrow D \cup D_i$

**end for**

Output: policy $\pi_{\text{MPC}}^z(s_t|\phi_\theta, f_\psi^z, R_\zeta^z)$

---

# B   PROOF OF THEOREM 1

*Proof.* Note that the $H$-step Q-function can be written as

$$Q_H^*(s, a) = \sum_{t=0}^{H-1} \gamma^t R(s_t, a_t^*)$$

where $s_0 = s$, $a_0^* = a$, and for $t > 0$

$$s_t = f(s_{t-1}, a_{t-1}^*) = f_t(s, a_{0:t-1}^*)$$
$$a_t^* = \pi^*(s_t)$$

Here $\pi^*$ denotes the optimal policy for the MPC problem equation 3. Because $Q_H^{*,z}(\phi_\theta(s), a)$ is the optimal Q-function for the latent MPC problem equation 7, we have

$$Q_H^{*,z}(\phi_\theta(s), a) \geq \sum_{t=0}^{H-1} \gamma^t R_\zeta^z(f_{\psi,t}^z(\phi_\theta(s), a_{0:t-1}^*), a_t^*)$$

Adding and subtracting $Q_H^*(s, a)$ we obtain

$$
\begin{aligned}
Q_H^{*,z}(\phi_\theta(s), a) \geq & Q_H^*(s, a) + \sum_{t=0}^{H-1} \gamma^t R_\zeta^z(f_{\psi,t}^z(\phi_\theta(s), a_{0:t-1}^*), a_t^*) - Q_H^*(s, a) \\
= & Q_H^*(s, a) + \sum_{t=0}^{H-1} \gamma^t (R_\zeta^z(f_{\psi,t}^z(\phi_\theta(s), a_{0:t-1}^*), a_t^*) - R(s_t, a_t^*)) \\
\geq & Q_H^*(s, a) - \sum_{t=0}^{H-1} ||\gamma^t (R_\zeta^z(f_{\psi,t}^z(\phi_\theta(s), a_{0:t-1}^*), a_t^*) - R(s_t, a_t^*))||_2 \\
= & Q_H^*(s, a) - (H\mathcal{L}_H)^{1/2} \geq Q_H^*(s, a) - H^{1/2}\varepsilon
\end{aligned}
\tag{10}
$$

Now let $\pi^{*,z}(z)$ be the optimal policy for the latent MPC problem in the latent space, then from the Q-function definition we have

$$Q_H^{z,*}(\phi_\theta(s), a) = \sum_{t=0}^{H-1} \gamma^t R_\zeta^z(s_t, a_t^{*,z})$$

where $z_0 = \phi_\theta(s)$, $a_0^{*,z} = a$, and for $t > 0$

$$z_t = f_\psi^z(s_{t-1}, a_{t-1}^{*,z}) = f_{\psi,t}^z(\phi_\theta(s), a_{0:k-1}^{*,z})$$
$$a_t^{*,z} = \pi^{*,z}(z_t)$$

The $H$-step Q-function of policy $\pi^{*,z} \circ \phi_\theta$ in problem equation 3 is then given by

$$Q_H^{\pi^{*,z} \circ \phi_\theta}(s,a) = \sum_{t=0}^{H-1} \gamma^t R(f_t(s, a_{0:t-1}^{*,z}), a_k^{*,z})$$

$$= Q_H^{*,z}(\phi(s), a) + \sum_{t=0}^{H-1} \gamma^t R(f_t(s, a_{0:t-1}^{*,z}), a_k^{*,z}) - Q_H^{*,z}(\phi(s), a)$$

$$= Q_H^{*,z}(\phi(s), a) + \sum_{t=0}^{H-1} \gamma^t (R(f_t(s, a_{0:t-1}^{*,z}), a_k^{*,z}) - R_\zeta^z(f_{\psi,t}^z(\phi_\theta(s), a_{0:t-1}^{*,z}), a_t^{*,z}))$$

$$\geq Q_H^{*,z}(\phi(s), a) - H^{1/2}\varepsilon \tag{11}$$

Now combining equation 10 and equation 11 we get

$$Q_H^{\pi^{*,z} \circ \phi_\theta}(s,a) \geq Q_H^*(s,a) - 2H^{1/2}\varepsilon$$

which means that $\pi^{*,z} \circ \phi$ is a $2H^{1/2}\epsilon$-optimal policy.   $\square$

## C  EXPERIMENTS WITH IMAGE OBSERVATIONS

To demonstrate that this method can solve tasks with rather high-dimensional state observations, we solve the single Pendulum task from two-gray-scale images ($2 \times 100 \times 100$ pixels). Similar to the Mutli-Pendulum experiments, we let the latent space be 3-dimensional. In provide two images per state to insure that velocity can be inferred and fully-observable assumption in maintained. We find that the prediction results are fairly similar to the kinematic state representation, but more noisy due to the course image observations. We train the model using only mostly offline, using only 1 iteration. We initialize the model with 10000 samples under random control trained for 100 epochs and then trained once more with another 10000 samples using the new control policy. This policy is able to solve the task efficiently (142 return) similarly to the kinematic state representation, producing similar prediction results. Its interesting to note that the open-loop model predicts that the pendulum will continue to oscillate and not stabilize. This could be evidence that the model over fits to the random training data, however it would also require very accurate model to predict stabilizing behavior far into the future.

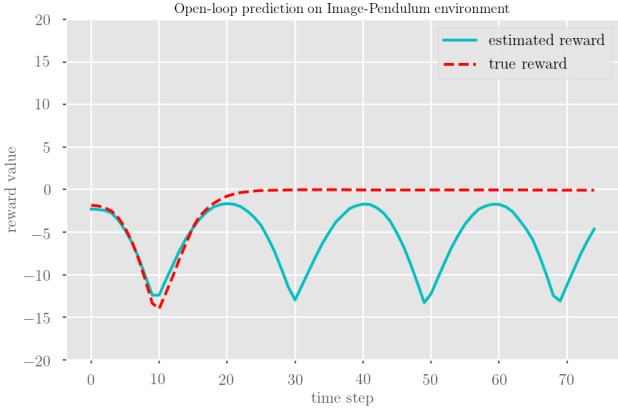

Figure 5: Depicted is the open loop prediction result of the Image-Pendulum environment successfully stabilized by the learned latent-reward CEM policy. The model is first trained on 10000 samples from a random policy, then another 10000 sample under the new control policy.

