# OpenReview forum: "Learning Latent State Spaces for Planning through Reward Prediction"
_ICLR.cc/2020/Conference — Reject_

### Official Review · AnonReviewer1 · 2019-10-12
**Official Blind Review #1**

**Rating:** 3

**Review:**

This paper claims that one only needs a reward prediction model to learn a good latent representation for model-based reinforcement learning. They introduce a method that learns a latent dynamics model exclusively from multi-step reward prediction, then use MPC to plan directly in the latent space. They claim this is sample efficient in the model-based way, and is more useful than predicting full states. They learn a model that predicts only current and future rewards conditioned on action sequences, and that observation reconstruction is unnecessary to learn a good latent space. They provide planning performance guarantees for approximate latent reward prediction models.

I tend to reject this work, because although I support the premise and believe it is very important, and like the style of experiments run with the use of distractors, I believe it is not impactful if only looking at the dense reward setting. The type of environment they describe that requires using lossy representations is also likely to only have sparse reward, so to only note in the conclusion as future work is not enough. The contributions consist only of learning a multi-step reward model for planning, and only provide results in two dense reward environments. In the second experiment with more difficult, high-dimensional observation and action space setting, two of the 3 baselines are left out, namely the state model and DeepMDP. I think it is crucial to include DeepMDP, as it is the one most likely to perform competitively with the proposed method.

The justification for Table 1 vs. Figure 3 are also very unclear, as to why SAC is trained with 10^6 samples while DeepMDP is trained under a random policy, whereas the original paper utilizes samples collected by the policy as its trains. SAC is off policy, and can therefore be evaluated with random data, rather than being used as an "upper bound baseline". The final evaluation performance in dashed line in Figure 3 also doesn't include standard deviation across the 5 seeds, which it should. The final results for SAC also do not match the performance in Figure 3, although it is hard to tell since the final performance in Table 1 is written in terms of number of environment steps while Figure 3 the axis is in terms of episodes.

Including sparse reward experiments would vastly help support the claims in the paper, as well as including the DeepMDP results for HalfCheetah and additional explanation of the difference in performance of SAC in Table 1 and Figure 3.

**Experience Assessment:**

I have read many papers in this area.

**Review Assessment: Checking Correctness Of Derivations And Theory:**

I assessed the sensibility of the derivations and theory.

**Review Assessment: Checking Correctness Of Experiments:**

I carefully checked the experiments.

**Review Assessment: Thoroughness In Paper Reading:**

I read the paper at least twice and used my best judgement in assessing the paper.

---

> ### Author Response · Authors · 2019-11-15
> **Response to reviewer #1**
>
> Thank you for your careful reviews,
>
> " I believe it is not impactful if only looking at the dense reward setting. The type of environment they describe that requires using lossy representations is also likely to only have sparse reward, so to only note in the conclusion as future work is not enough"
>
> By focusing only on rewards, the sparse reward setting is certainly a weakness of our method compared with the state-reconstruction counterparts. Nevertheless, the reward signals collected by our method is exactly the same as any model-free algorithm. This means that our method might be able to tackle sparse reward problems by adopting some exploration ideas from model-free algorithms. Ideally the current method would still work given that the reward prediction horizon is sufficiently long to observe a non-zero reward, or else the representation may just collapse most states to the same latent state. There may be some trade off between state reconstruction and reward prediction necessary for a sense of "observability" in the finite horizon case.
>
> " I think it is crucial to include DeepMDP, as it is the one most likely to perform competitively with the proposed method"
>
> We agree that including DeepMDP results for Half-cheetah is important, which has been updated. The additional results show that the DeepMDP performance is half of that of the reward prediction model with 1-cheetach, and the performance gap is further increased to more than 10 times in the 5-cheetach environment.
>
> "SAC is off policy, and can therefore be evaluated with random data, rather than being used as an "upper bound baseline"
>
> Thank you for this comment, will aim to more clearly describe the purpose of each figure. Since table 1 shows SAC being trained on 1e6 on-policy samples and other methods are being trained on only 2e4 off-policy samples, we are almost certain that the result would only be more favorable in comparison to train SAC off-policy. We chose to use SAC as a baseline here only to provide a reasonable reference of what state of the art performance would be on this task, since the multi-pendulum is a somewhat novel environment.
>
> "DeepMDP is trained under a random policy, whereas the original paper utilizes samples collected by the policy as its trains"
>
> We acknowledge that in table 1 all algorithms except for SAC were trained offline, however figure 3 compares all algorithms in an on-policy setting. For this reason, we decided to keep Deepmdp consistent with other algorithms in Table 1. In the original DeepMDP paper, the “donut world” experiments were performed similarly under exhaustive sampling of the state-space, not based on a policy.
>
> "The final results for SAC also do not match the performance in Figure 3"
>
> In figure 3 i.e. the on-policy setting, SAC was trained with far fewer samples over 600 episodes which is only about 24000 samples compared to 1e6 samples used in table 1.
>
> "Final performance should have standard-deviation"
>
> Thank you for pointing this out, this would make the results more informative . We will update the final plots to display shaded 1-standard deviation regions for all final performances.
>
> Thank you

---

### Official Review · AnonReviewer2 · 2019-10-21
**Official Blind Review #2**

**Rating:** 3

**Review:**

This paper presents a technique for model based RL/planning with latent dynamics models, which learns the latent model only using reward prediction. This is in contrast to existing work which generally use a combination of reward prediction and state reconstruction to learn the latent model. The paper suggests that by removing the state reconstruction loss, the agent can learn to ignore irrelevant parts of the state, which should enable better performance in settings where state reconstruction is challenging.

Overall the motivation for this work is good, and the idea is promising. Difficulty in reconstructing high dimensional states is a challenge for learning latent dynamics models. The paper is also very well written and easy to follow.

My concerns are centered around the experimental evaluation. Specifically, I see the following issues: (1) the experimental environments seem artificial, and hand tailored for this method, (2) given that the proposed method is a minor modification to the PlaNet paper, it seems that PlaNet should be included as a comparison (especially because it has been shown to work on high dimensional states), and (3) the proposed method seems very prone to overfitting to the given task, and there should be an analysis of how the proposed change affects generalization and robustness.

(1): The testing environments contain many distractor pendulums/cheetahs, which makes state reconstruction especially challenging. While this does seem to be the point the authors are trying to show, the environments are an extreme, almost artificial, case of difficult state reconstruction. Would the same results hold in more realistic settings, for example, visual robot manipulation in a cluttered scene? Model based RL with video prediction models has been shown to work in such real cluttered robot manipulation environments. Showing that the proposed method can outperform such approaches in robot manipulation settings would be a powerful result. The results on images in the appendix seem to show a delta between the true and predicted reward, suggesting that the proposed method does not yet work on images. Why might this be the case?

(2): From what I can see, the proposed method is very similar to the PlaNet algorithm with state reconstruction loss removed. Given the similarity, PlaNet should be included as a comparison in both the pendulum and cheetah environments. Similarly why was DeepMDP performance not shown in the Cheetah environment?

(3): One of the strengths of model based reinforcement learning is the ability to plan to reach unseen goals with a model trained via self-supervision or different goals. Does the proposed approach lose some of this, by overfitting to only the task reward? I suspect that in generalizing to unseen tasks, a model trained with state prediction would potentially perform much better. If trained on many tasks, could this method achieve similar generalization?

Due to some of these questions which remain unanswered by the experimental evaluation my current rating is Weak Reject. If the authors are able to clarify some of the questions above I may adjust my score.


**Experience Assessment:**

I have published one or two papers in this area.

**Review Assessment: Checking Correctness Of Derivations And Theory:**

I assessed the sensibility of the derivations and theory.

**Review Assessment: Checking Correctness Of Experiments:**

I carefully checked the experiments.

**Review Assessment: Thoroughness In Paper Reading:**

I read the paper thoroughly.

---

> ### Author Response · Authors · 2019-11-15
> **Response to reviewer #2**
>
> Thank you for your careful reviews,
>
> "The testing environments contain many distractor pendulums/cheetahs, which makes state reconstruction especially challenging. While this does seem to be the point the authors are trying to show, the environments are an extreme, almost artificial, case of difficult state reconstruction."
>
> It is true that the experiments are intentionally designed to investigate and emphasize the desirable properties of the reward-prediction method. We agree that a more grounded example such as a vision-based grasping task in a cluttered environment would be very convincing and we should surely show this in the future.
>
> "The results on images in the appendix seem to show a delta between the true and predicted reward, suggesting that the proposed method does not yet work on images. Why might this be the case?"
>
> Thank you for bringing this to attention. We did not sufficiently explain this result in context to the main results. Preliminary results show that the method works for images, but have not been thoroughly benchmarked yet. The figure shows a single open-loop prediction of reward in the pendulum environment from images. The open-loop prediction is not perfect and will accumulate error after about 20-steps, especially for predicting a stabilizing behavior. However, notice that the red line is the true reward and is stabilized under the MPC controller which is replanning based on the true observation at every time step, not open-loop.
>
> "From what I can see, the proposed method is very similar to the PlaNet algorithm with state reconstruction loss removed. Given the similarity, PlaNet should be included as a comparison in both the pendulum and cheetah environments. Similarly why was DeepMDP performance not shown in the Cheetah environment?"
>
> We agree that PlaNet has a similar framework for latent state-space learning. However, the latent model of PlaNet consists of both stochastic and deterministic components and a variational objective. These key differences make it difficult to have a fair comparison between PlaNet and the proposed method. The benefit of reward prediction loss over state reconstruction losses can be clearly observed in the experiments for the state model and the reward model.
> For DeepMDP, we agree that including DeepMDP results for multi-cheetah is important and we have updated the paper immediately. The additional results show that the DeepMDP performance is half of that of the reward prediction model with 1-cheetah, and the performance gap is further increased to more than 10 times in the 5-cheetah environment.
>
> "One of the strengths of model based reinforcement learning is the ability to plan to reach unseen goals with a model trained via self-supervision or different goals. Does the proposed approach lose some of this, by overfitting to only the task reward?"
>
> This is an interesting point and we agree that there is potential work to be done to investigate on some kind of meta-learning task. You are correct that this reward prediction module would be specific to a particular task reward. However, for a similar task like reaching an unseen goal, the encoding and forward dynamics function can certainly be reused in planning while only the reward function requires to be re-learned. We think that it is promising to pose the multi-task setting as a proper meta-learning problem where we learn the representation over a task distribution. This time we chose to focus on the single task setting as a proof of concept.
>
> Thank you.

---

### Official Review · AnonReviewer3 · 2019-10-24
**Official Blind Review #3**

**Rating:** 6

**Review:**

Summary:
This paper proposes a novel algorithm for planning on specific domains through latent reward prediction. The proposed model uses an encoder to learn embedding the state to the latent state, a forward dynamics function to learn dynamical system in latent state space, and a reward function to estimate the reward given a latent state and an action. Using these functions, the authors define the objective using the mean-squared error between true and multi-step prediction of rewards. To justify the proposed method, the authors provide a theoretical analysis and experimental results on specific RL domains, multi-pendulum and multi-cheetah, which contain irrelevant aspects of the state.

Comments:
This paper is well-written and easy to understand.
- In this paper, the authors assume deterministic transition and use deterministic function for latent transition. It seems to be the authors want to use MPC, which is a powerful planning algorithm. However, many RL tasks are modeled with stochastic transition. In stochastic transition cases, is the proposed algorithm still valid?
- As shown in Figure 3, even proposed method shows better performance than SAC in early episode but table 1 says that SAC shows the best convergence results in any number of pendulums except the single pendulum case. It seems to be different results from intuition, because the authors emphasize that the strength of the proposed method is efficiency of learning in RL tasks with irrelevant information.

Questions and minor comments:
- What objective is used to learn the latent model of the state-prediction model algorithm?
- Providing detailed experimental settings, like detailed settings for three deterministic feed-forward neural networks, and results such as consumed CPU time will help the comparison algorithms.

**Experience Assessment:**

I have read many papers in this area.

**Review Assessment: Checking Correctness Of Derivations And Theory:**

I assessed the sensibility of the derivations and theory.

**Review Assessment: Checking Correctness Of Experiments:**

I assessed the sensibility of the experiments.

**Review Assessment: Thoroughness In Paper Reading:**

I made a quick assessment of this paper.

---

> ### Author Response · Authors · 2019-11-15
> **Response to reviewer #3**
>
> Thank you for your careful reviews,
>
> "In this paper, the authors assume deterministic transition and use deterministic function for latent transition. It seems to be the authors want to use MPC, which is a powerful planning algorithm. However, many RL tasks are modeled with stochastic transition. In stochastic transition cases, is the proposed algorithm still valid?"
>
> We agree that, in the future, an explicit representation of uncertainty and stochasticity is necessary for state-of-art application, although most benchmark tasks, including the ones considered in this paper, are purely deterministic. When it comes to stochastic environments, It is possible to extend the latent reward prediction model to include stochastic components. The cross-entropy method (CEM) used for MPC in this paper naturally extends to a stochastic setting.
>
> "As shown in Figure 3, even the proposed method shows better performance than SAC in early episode but table 1 says that SAC shows the best convergence results in any number of pendulums except the single pendulum case. It seems to be different results from intuition, because the authors emphasize that the strength of the proposed method is efficiency of learning in RL tasks with irrelevant information. "
>
> In the current state of model-based RL, it is widely observed that model-free algorithms generally perform better in the limit of samples while our goal is to provide a sample-efficient model-based algorithm that scales well to high-dimensional observations with irrelevant information. Table 1 is meant to be an ablation study for the multi-pendulum environment, where SAC as a performance baseline. The other methods in Table 1 consumed only  1/50 of the samples as SAC. in Wang et al., 2019 [https://arxiv.org/abs/1907.02057]
>
> "What objective is used to learn the latent model of the state-prediction model algorithm?", "Providing detailed experimental settings"
>
> Thank you for this clarifying question. Similar to the reward-only model, the state-prediction model has a multi-step mean squared error loss on full-observation prediction as well as reward prediction. We will add this information and formula along with architecture and algorithmic details to the appendix.
>
> Thank you.

---

### Author Response · Authors · 2019-11-15
**General comment and updates**

We would like to thank the reviewers for their helpful comments and feedback. We were truly appreciative to see that the problem we are addressing is well-received with several comments on how to proceed further in this domain.

We have addressed the general concern of including Deepmdp results in figure 4 (1-cheetah and 5-cheetah), as it is a competitive baseline in this setting and may inform the effect of prediction horizon in our method (Deepmdp's 1-step vs our multi-step method). We have also made a slight improvement to the theoretical performance bound in the theorem statement and appendix. Both of these changes have been updated in the submission.

Thank you.

---

### Decision · Program_Chairs · 2019-12-19

**Decision:**

Reject

**Comment:**

The authors propose a model-based RL algorithm, consisting of learning a
deterministic multi-step reward prediction model and a vanilla CEM-based MPC
actor.
In contrast to prior work, the model does not attempt to learn from observations
nor is a value function learned.
The approach is tested on task from the mujoco control suit.

The paper is below acceptance threshold.
It is a variation on previous work form Hafner et al.
Furthermore, I think the approach is fundamentally limited: All the learning
derives from the immediate, dense reward signal, whereas the main challenges in RL
are found in sparse reward settings that require planning over long horizons, where value
functions or similar methods to assign credit over long time windows are
absolutely essential.